# DiP-GNN: Discriminative Pre-Training of Graph Neural Networks

## Abstract

Graph neural network (GNN) pre-training methods have been proposed to en-
hance the power of GNNs. Specifically, a GNN is first pre-trained on a large-scale
unlabeled graph and then fine-tuned on a separate small labeled graph for down-
stream applications, such as node classification. One popular pre-training method
is to mask out a proportion of the edges, and a GNN is trained to recover them.
However, such a generative method suffers from graph mismatch. That is, the
masked graph input to the GNN deviates from the original graph. To alleviate
this issue, we propose DiP-GNN (Discriminative Pre-training of Graph Neural
Networks). Specifically, we train a generator to recover identities of the masked
edges, and simultaneously, we train a discriminator to distinguish the generated
edges from the original graph's edges. The discriminator is subsequently used
for downstream fine-tuning. In our pre-training framework, the graph seen by the
discriminator better matches the original graph because the generator can recover
a proportion of the masked edges. Extensive experiments on large-scale homo-
geneous and heterogeneous graphs demonstrate the effectiveness of the proposed
framework. Our code will be publicly available.

## 1    Introduction

Graph neural networks (GNNs) have achieved superior performance in various applications, such as
node classification (Kipf & Welling, 2017), knowledge graph modeling (Schlichtkrull et al., 2018)
and recommendation systems (Ying et al., 2018). To enhance the power of GNNs, generative pre-
training methods are developed (Hu et al., 2020b). During the pre-training stage, a GNN incor-
porates topological information by training on a large-scale unlabeled graph in a self-supervised
manner. Then, the pre-trained model is fine-tuned on a separate small labeled graph for downstream
applications. Generative GNN pre-training is akin to masked language modeling in language model
pre-training (Devlin et al., 2019). That is, for an input graph, we first randomly mask out a propor-
tion of the edges, and then a GNN is trained to recover the original identity of the masked edges.

One major drawback with the abovementioned approach is *graph mismatch*. That is, the input graph
to the GNN deviates from the original one since a considerable amount of edges are dropped. This
causes changes in topological information, e.g., node connectivity. Consequently, the learned node
embeddings may not be desirable.

To mitigate the above issues, we propose DiP-GNN ( **Di**scriminative **P**re-training of **G**raph **N**eural
**N**etworks). In DiP-GNN, we simultaneously train a generator and a discriminator. The generator
is trained similar to existing generative pre-training approaches, where the model seeks to recover
the masked edges and outputs a reconstructed graph. Subsequently, the reconstructed graph is fed to
the discriminator, which predicts whether each edge resides in the original graph (i.e., a true edge)
or is wrongly constructed by the generator (i.e., a fake edge). After pre-training, we fine-tune the
discriminator on downstream tasks. Figure 1 illustrates our training framework. Note that our work
is related to Generative Adversarial Nets (GAN, Goodfellow et al. 2014), and detailed discussions
are presented in Section 3.4. We remark that similar approaches have been used in natural language
processing (Clark et al., 2020). However, we identify the graph mismatch problem (see Section 4.5),
which is specific to graph-related applications and is not observed in natural language processing.

The proposed framework is more advantageous than generative pre-training. This is because the
reconstructed graph fed to the discriminator better matches the original graph compared with the

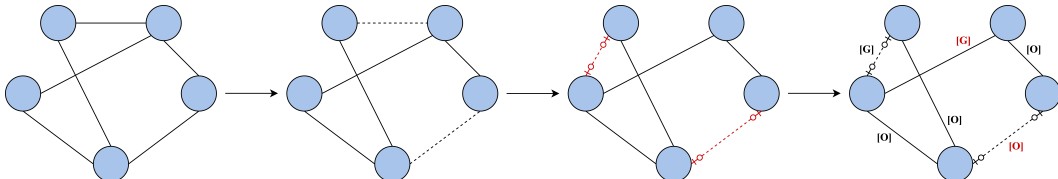

Figure 1: Illustration of DiP-GNN. From left to right: Original graph; Graph with two masked edges (dashed lines); Reconstructed graph created by the generator (generated edges are the dashed red lines); Discriminator labels each edge as *[G]* (generated) or *[O]* (original), where there are two wrong labels (shown in red).

masked graph fed to the generator. Consequently, the discriminator can learn better node embeddings. Such a better alignment is because the generator recovers the masked edges during pre-training, i.e., we observe that nearly 40% of the missing edges can be recovered. We remark that in our framework, the graph fed to the generator has missing edges, while the graph fed to the discriminator contains wrong edges since the generator may make erroneous predictions. However, empirically we find that missing edges hurt more than wrong ones, making discriminative pre-training more desirable (see Section 4.5 in the experiments).

We demonstrate effectiveness of DiP-GNN on large-scale homogeneous and heterogeneous graphs. Results show that the proposed method significantly outperforms existing generative pre-training and self-supervised learning approaches. For example, on the homogeneous Reddit dataset (Hamilton et al., 2017) that contains 230k nodes, we obtain an improvement of 1.1 in terms of F1 score; and on the heterogeneous OAG-CS graph (Tang et al., 2008) that contains 1.1M nodes, we obtain an improvement of 2.8 in terms of MRR score in the paper field prediction task.

## 2 BACKGROUND

⋄ **Graph Neural Networks**. Graph neural networks compute a node's representation by aggregating information from the node's neighbors. Concretely, for a multi-layer GNN, the feature vector $h_v^{(k)}$ of node $v$ at the $k$-th layer is

$$a_v^{(k)} = \text{Aggregate}\left(\left\{h_u^{(k-1)} \, \forall u \in \text{Neighbor}(v)\right\}\right), \; h_v^{(k)} = \text{Combine}\left(a_v^{(k)}, h_v^{(k-1)}\right),$$

where $\text{Neighbor}(v)$ denotes all the neighbor nodes of $v$. Various implementations of $\text{Aggregate}(\cdot)$ and $\text{Combine}(\cdot)$ are proposed for both homogeneous (Defferrard et al., 2016; Kipf & Welling, 2017; Velickovic et al., 2018; Xu et al., 2019) and heterogeneous graphs (Schlichtkrull et al., 2018; Wang et al., 2019; Zhang et al., 2019; Hu et al., 2020c).

⋄ **Graph Neural Network Pre-Training**. Previous unsupervised learning methods leverage the graph's proximity (Tang et al., 2015) or information gathered by random walks (Perozzi et al., 2014; Grover & Leskovec, 2016; Dong et al., 2017; Qiu et al., 2018). However, the learned embeddings cannot be transferred to unseen nodes, limiting the methods' applicability. Other unsupervised learning algorithms adopt contrastive learning (Hassani & Ahmadi, 2020; Qiu et al., 2020; Zhu et al., 2020; 2021; You et al., 2020; 2021). That is, we generate two views of the same graph, and then maximize agreement of node presentations in the two views. However, our experiments reveal that these methods do not scale well to extremely large graphs with millions of nodes.

Many GNN pre-training methods focus on generative objectives. For example, GAE (Graph Auto-Encoder, Kipf & Welling 2016) proposes to reconstruct the graph structure; GraphSAGE (Hamilton et al., 2017) optimizes an unsupervised loss derived from a random-walk-based metric; and DGI (Deep Graph Infomax, Velickovic et al. 2019) maximizes the mutual information between node representations and a graph summary representation.

There are also pre-training methods that extract graph-level representations, i.e., models are trained on a large amount of small graphs instead of a single large graph. For example, Hu et al. 2020a propose pre-training methods that operate on both graph and node level; and InfoGraph (Sun et al., 2020) proposes to maximize the mutual information between graph representations and representations of the graphs' sub-structures. In this work, we focus on pre-training GNNs on a single large graph instead of multiple small graphs.

# 3 METHOD

We formally introduce the proposed discriminative GNN pre-training framework DiP-GNN. The algorithm contains two ingredients that operate on edges and features.

## 3.1 EDGE GENERATION AND DISCRIMINATION

Suppose we have a graph $\mathcal{G} = (\mathcal{N}, \mathcal{E})$, where $\mathcal{N}$ denotes all the nodes and $\mathcal{E}$ denotes all the edges. We randomly mask out a proportion of the edges, such that $\mathcal{E} = \mathcal{E}_u \cup \mathcal{E}_m$, where $\mathcal{E}_u$ is the unmasked set of edges and $\mathcal{E}_m$ is the set of edges that are masked out.

For a masked edge $e = (n_1, n_2) \in \mathcal{E}_m$, where $n_1$ and $n_2$ are the two nodes connected by $e$, the generator's goal is to predict $n_1$ given $n_2$ and the unmasked edges $\mathcal{E}_u$. For each node $n$, we compute its representation $h_g(n) = f_g^e(n, \theta_g^e)$ using the generator $f_g^e(\cdot, \theta_g^e)$, which is parameterized by $\theta_g^e$. We remark that the computation of $h_g(\cdot)$ only relies on the unmasked edges $\mathcal{E}_u$. We assume that the generation process of each edge is independent. Then, we have the prediction probability

$$p(n_1|n_2, \mathcal{E}_u) = \frac{\exp\left(d(h_g(n_1), h_g(n_2))\right)}{\sum_{n' \in \mathcal{C}} \exp\left(d(h_g(n'), h_g(n_2))\right)}, \quad \mathcal{C} = \{n_1\} \cup (\mathcal{N} \setminus \text{Neighbor}(n_2)). \quad (1)$$

Here, $\mathcal{C}$ is the candidate set for $n_1$, which contains all the nodes that are not connected to $n_2$ except $n_1$ itself. Moreover, the distance function $d(\cdot, \cdot)$ is chosen as a trainable cosine similarity, i.e.,

$$d(u, v) = \frac{(W^{\cos}u)^\top v}{||W^{\cos}u|| \cdot ||v||}, \quad (2)$$

where $W^{\cos}$ is a trainable weight. The training loss for the generator is defined as

$$\mathcal{L}_g^e(\theta_g^e) = \sum_{(n_1, n_2) \in \mathcal{E}_m} -\log p(n_1|n_2, \mathcal{E}_u), \quad (3)$$

which is equivalent to maximizing the likelihood of correct predictions.

The goal of the generator is to recover the masked edges in $\mathcal{E}_m$. Therefore, after we train the generator, we use the trained model to generate $\mathcal{E}_g = \{(\widehat{n}_1, n_2)\}_{(n_1, n_2) \in \mathcal{E}_m}$, where each $\widehat{n}_1$ is the model's prediction as $\widehat{n}_1 = \arg\max_{n' \in \mathcal{C}} p(n'|n_2, \mathcal{E}_u)$. Because the generator cannot correctly predict every edge, some edges in $\mathcal{E}_g$ are wrongly generated (i.e., not in $\mathcal{E}_m$). We refer to such edges as *fake* edges, and the rest as *true* edges. Concretely, we denote the true edges $\mathcal{E}_{\text{true}} = \mathcal{E}_u \cup (\mathcal{E}_m \cap \mathcal{E}_g)$, i.e., the unmasked edges and the edges correctly generated by the generator. Correspondingly, we denote the fake edges $\mathcal{E}_{\text{fake}} = \mathcal{E} \setminus \mathcal{E}_{\text{true}}$.

The discriminator is trained to distinguish edges that are from the original graph (i.e., the true edges) and edges that are not (i.e., fake edges). Specifically, given the true edges $\mathcal{E}_{\text{true}}$ and the fake ones $\mathcal{E}_{\text{fake}}$, we first compute $h_d(n) = f_d^e(n, \theta_d^e)$ for every node $n \in \mathcal{N}$, where $f_d^e(\cdot, \theta_d^e)$ is the discriminator model parameterized by $\theta_d^e$. We highlight that different from computing $h_g(\cdot)$, the computation of $h_d(\cdot)$ relies on all the edges, such that the discriminator can separate a fake edge from a true one. Then, for each edge $e = (n_1, n_2) \in \mathcal{E}_{\text{true}} \cup \mathcal{E}_{\text{fake}}$, the discriminator outputs

$$p_{\text{fake}} = p(e \in \mathcal{E}_{\text{fake}}|\mathcal{E}_{\text{true}}, \mathcal{E}_{\text{fake}}) = \text{sigmoid}\left(d(h_d(n_1), h_d(n_2))\right), \quad (4)$$

where $d(\cdot, \cdot)$ is the distance function in Eq. 2. The training loss for the discriminator is the binary cross-entropy loss of predicting whether an edge is fake or not, defined as

$$\mathcal{L}_d^e(\theta_d^e) = \sum_{e \in \mathcal{E}_{\text{true}} \cup \mathcal{E}_{\text{fake}}} -\mathbf{1}\{e \in \mathcal{E}_{\text{fake}}\} \log(p_{\text{fake}}) - \mathbf{1}\{e \in \mathcal{E}_{\text{true}}\} \log(1 - p_{\text{fake}}), \quad (5)$$

where $\mathbf{1}\{\cdot\}$ is the indicator function.

The edge loss is the weighted sum of the generator's and the discriminator's loss

$$\mathcal{L}^e(\theta_g^e, \theta_d^e) = \mathcal{L}_g^e(\theta_g^e) + \lambda \mathcal{L}_d^e(\theta_d^e), \quad (6)$$

where $\lambda$ is a hyper-parameter. Note that structures of the generator $f_g^e$ and the discriminator $f_d^e$ are flexible, e.g., they can be graph convolutional networks (GCN) or graph attention networks (GAT).

## 3.2 FEATURE GENERATION AND DISCRIMINATION

In real-world applications, nodes are often associated with features. For example, in the Reddit dataset (Hamilton et al., 2017), a node's feature is a vectorized representation of the post corresponding to the node. As another example, in citation networks (Tang et al., 2008), a paper's title

can be treated as a node's feature. Previous work (Hu et al., 2020b) has demonstrated that generating features and edges simultaneously can improve the GNN's representation power.

Node features can be either texts (e.g., in citation networks) or vectors (e.g., in recommendation systems). In this section, we develop feature generation and discrimination procedures for texts. Vector features are akin to encoded text features, and we can use linear layers to generate and discriminate them. Details about vector features are deferred to Appendix B.

For text features, we parameterize both the feature generator and the feature discriminator using bi-directional Transformer models (Vaswani et al., 2017), similar to BERT (Devlin et al., 2019). Denote $f_g^f(\cdot, \theta_g^f) = \mathrm{trm}_g \circ \mathrm{emb}_g(\cdot)$ the generator parameterized by $\theta_g^f$, where $\mathrm{emb}_g$ is the word embedding function and $\mathrm{trm}_g$ denotes subsequent Transformer layers. For an input text feature $\mathbf{x} = [x_1, \cdots, x_L]$ where $L$ is the sequence length, we randomly select indices to mask out, i.e., we randomly select an index set $\mathcal{M} \subset \{1, \cdots, L\}$. For a masked position $i \in \mathcal{M}$, the prediction probability is given by

$$p(x_i|\mathbf{x}) = \frac{\exp\left(\mathrm{emb}_g(x_i)^\top v_g(x_i)\right)}{\sum_{x' \in \mathrm{vocab}} \exp\left(\mathrm{emb}_g(x')^\top v_g(x')\right)}, v_g(x_i) = \mathrm{trm}_g\left(W_g^{\mathrm{proj}}\left[h_g(n_\mathbf{x}), \mathrm{emb}_g(x_i)\right]\right). \quad (7)$$

Here $W_g^{\mathrm{proj}}$ is a trainable weight and $h_g(n_\mathbf{x})$ is the representation of the node corresponding to $\mathbf{x}$ computed by the edge generation GNN. Note that we concatenate the text embedding $\mathrm{emb}_g(x_i)$ and the feature node's embedding $h_g(n_\mathbf{x})$, such that the feature generator can aggregate information from the graph structure. We train the generator by maximizing the probability of predicting the correct token, i.e., by minimizing the loss

$$\mathcal{L}_g^f(\theta_g^e, \theta_g^f) = \sum_\mathbf{x} \sum_{i \in \mathcal{M}} -\log p(x_i|\mathbf{x}). \quad (8)$$

After we train the generator, we use the trained model to predict all the masked tokens, after which we obtain a new text feature $\mathbf{x}^{\mathrm{corr}}$. Here, we set $x_i^{\mathrm{corr}} = x_i$ for $i \notin \mathcal{M}$ and $x_i^{\mathrm{corr}} = \widehat{x}_i$ for $i \in \mathcal{M}$, where $\widehat{x}_i = \mathrm{argmax}_{x' \in \mathrm{vocab}} p(x_i|\mathbf{x})$ is the generator's prediction.

The discriminator is trained to distinguish the fake tokens (i.e., wrongly generated tokens) from the true ones (i.e., the unmasked and correctly generated tokens) in $\mathbf{x}^{\mathrm{corr}}$. Similar to the generator, we denote $f_d^f(\cdot, \theta_d^f) = \mathrm{trm}_d \circ \mathrm{emb}_d(\cdot)$ as the discriminator parameterized by $\theta_d^f$. For each position $i$, the discriminator's prediction probability is defined as

$$p(x_i^{\mathrm{corr}} = x_i) = \mathrm{sigmoid}\left(w^\top v_d(x_i^{\mathrm{corr}})\right), \ v_d(x_i^{\mathrm{corr}}) = \mathrm{trm}_d\left(W_d^{\mathrm{proj}}\left[h_d(n_\mathbf{x}), \mathrm{emb}_d(x_i^{\mathrm{corr}})\right]\right). \quad (9)$$

Here $w$ and $W_d^{\mathrm{proj}}$ are trainable weights and $h_d(n_\mathbf{x})$ is the representation of the node corresponding to $\mathbf{x}$ computed by the edge discriminator GNN. The training loss for the discriminator is

$$\mathcal{L}_d^f(\theta_d^e, \theta_d^f) = \sum_\mathbf{x} \sum_{i=1}^L -\mathbf{1}\{x_i^{\mathrm{corr}} = x_i\} \log(p_{\mathrm{true}}) - \mathbf{1}\{x_i^{\mathrm{corr}} \neq x_i\} \log(1 - p_{\mathrm{true}}), \quad (10)$$

where $p_{\mathrm{true}} = p(x_i^{\mathrm{corr}} = x_i)$ and $\mathbf{1}\{\cdot\}$ is the indicator function.

The text feature loss is defined as

$$\mathcal{L}^f(\theta_g^e, \theta_g^f, \theta_d^e, \theta_d^f) = \mathcal{L}_g^f(\theta_g^e, \theta_g^f) + \lambda \mathcal{L}_d^f(\theta_d^e, \theta_d^f), \quad (11)$$

where $\lambda$ is a hyper-parameter.

## 3.3 Model Training

We jointly minimize the edge loss and the feature loss, where the loss function is

$$\begin{aligned} \mathcal{L}(\theta_g^e, \theta_g^f, \theta_d^e, \theta_d^f) &= \mathcal{L}^e(\theta_g^e, \theta_d^e) + \mathcal{L}^f(\theta_g^e, \theta_g^f, \theta_d^e, \theta_d^f) \\ &= \left(\mathcal{L}_g^e(\theta_g^e) + \mathcal{L}_g^f(\theta_g^e, \theta_g^f)\right) + \lambda\left(\mathcal{L}_d^e(\theta_d^e) + \mathcal{L}_d^f(\theta_d^e, \theta_d^f)\right). \end{aligned} \quad (12)$$

Here, $\lambda$ is the weight of the discriminator's loss. We remark that our framework is flexible because the generator's loss ($\mathcal{L}_g^e$ and $\mathcal{L}_g^f$) is decoupled from the discriminator's ($\mathcal{L}_d^e$ and $\mathcal{L}_d^f$). As such, existing generative pre-training methods can be applied to train the generator. In DiP-GNN, the discriminator has a better quality than the generator because of the graph mismatch issue (see Section 4.5). Therefore, after pre-training, we discard the generator and fine-tune the *discriminator* on downstream tasks. A detailed training pipeline is presented in Appendix A.

### 3.4 COMPARISON WITH GAN

We remark that our framework is different from Generative Adversarial Nets (GAN, Goodfellow et al. 2014). In GAN, the generator-discriminator training framework is formulated as a min-max game, where the generator is trained adversarially to fool the discriminator. The two models are updated using alternating gradient descent/ascent.

However, the min-max game formulation of GAN is not applicable to our framework. This is because in GNN pre-trianing, the generator generates discrete edges, unlike continuous pixel values in the image domain. Such a property prohibits back-propagation from the discriminator to the generator. Existing works (Wang et al., 2018) use reinforcement learning (specifically policy gradient) to circumvent the non-differentiability issue. However, reinforcement learning introduces extensive hyper-parameter tuning and suffers from scalability issues. For example, the largest graph used in Wang et al. 2018 only contains 18k nodes, whereas the smallest graph used in our experiments has about 233k nodes.

Additionally, the goal of GAN is to train good-quality generators, which is different from our focus. In our discriminative pre-training framework, we focus on the discriminator because of better graph alignments. In practice, we find that accuracy of the generator is already high even without the discriminator, e.g., the accuracy is higher than 40% with 255 negative samples. And we observe that further improving the generator does not benefit downstream tasks.

## 4 EXPERIMENTS

We implement all the algorithms using PyTorch (Paszke et al., 2019) and PyTorch Geometric (Fey & Lenssen, 2019). Experiments are conducted on NVIDIA A100 GPUs. By default, we use Heterogeneous Graph Transformer (HGT, Hu et al. 2020c) as the backbone GNN. We also discuss other choices in the experiments. Training and implementation details are deferred to Appendix C.

### 4.1 SETTINGS AND DATASETS

⋄ **Settings.** We consider a *node transfer* setting in the experiments. In practice we often work with a single large-scale graph, on which labels are sparse. In this case, we can use the large amount of unlabeled data as the pre-training dataset, and the rest are treated as labeled fine-tuning nodes. Correspondingly, edges between pre-training nodes are added to the pre-training data, and edges between fine-tuning nodes are added to the fine-tuning data. In this way, the model cannot see the fine-tuning data during pre-training, and vice versa.

We remark that our setting is different from conventional self-supervised learning settings, namely we pre-train and fine-tune on two separate graphs. This meets the practical need of transfer learning, e.g., a trained GNN needs to transfer across locales and time spans in recommendation systems.

⋄ **Homogeneous Graph.** We use the Reddit dataset (Hamilton et al., 2017), which is a publicly available large-scale graph. In this graph, each node corresponds to a post, and is labeled with a "subreddit". Each node has a 603-dimensional feature vector constructed from the corresponding post. Two nodes (posts) are connected if the same user commented on both. The dataset contains posts from 50 subreddits sampled from posts initiated in September 2014. In total, there are 232,965 posts with an average node degree of 492. We use 70% of the data as the pre-training data, and the rest as the fine-tuning data, which are further split into training, validation, and test sets equally. We consider node classification as the downstream fine-tuning task.

⋄ **Product Recommendation Graph.** We collect in-house product recommendation data from an e-commerce website. We build a bi-partite graph with two node types: search queries and product ids. The dataset contains about 633k query nodes, 2.71M product nodes, and 228M edges. We sample 70% of the nodes (and corresponding edges) for pre-training, and the rest are evenly split for fine-tuning training, validation and testing. We consider link prediction as the downstream task, where for each validation and test query node, we randomly mask out 20% of its edges to recover. For each masked edge that corresponds to a query node and a positive product node, we randomly sample 255 negative products. The task is to find the positive product out of the total 256 products.

◇ **Heterogeneous Graph.** We use the OAG-CS dataset (Tang et al., 2008; Sinha et al., 2015), which is a publicly available heterogeneous graph containing computer science papers. The dataset contains over 1.1M nodes and 28.4M edges. In this graph, there are five node types (institute, author, venue, paper and field) and ten edge types. The "field" nodes are further categorized into six levels from L0 to L5, which are organized using a hierarchical tree. Details are shown in Figure 2.

We use papers published before 2014 as the pre-training dataset (63%), papers published between 2014 (inclusive) and 2016 (inclusive) as the fine-tuning training set (20%), papers published in 2017 as the fine-tuning validation set (7%), and papers published after 2017 as the fine-tuning test set (10%). During fine-tuning, by default we only use 10% of the fine-tuning training data (i.e., 2% of the overall data) because in practice labeled data are often scarce. We consider three tasks for fine-tuning: author name disambiguation (AD), paper field classification (PF) and paper venue classification (PV).

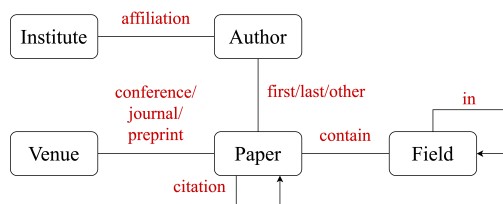

Figure 2: Details of OAG-CS. There are 5 node types (in black) and 10 edge types (in red).

For paper field classification, we only consider L2 fields. In the experiments, we use the pre-processed graph from Hu et al. 2020b.

## 4.2 IMPLEMENTATION DETAILS

◇ **Graph subsampling.** In practice, graphs are often too large to fit in the hardware, e.g., the Reddit dataset (Hamilton et al., 2017) contains over 230k nodes. Therefore, we sample a dense subgraph from the large-scale graph in each training iteration. For homogeneous graphs, we apply the LADIES algorithm (Zou et al., 2019), which theoretically guarantees that the sampled nodes are highly inter-connected with each other and can maximally preserve the graph structure. For heterogeneous graphs, we use the HGSampling algorithm (Hu et al., 2020b), which is a heterogeneous version of LADIES.

◇ **Node sampling for the edge generator.** In the edge generator, for a masked edge $(s, t)$, we fix the node $t$ and seek to identify the other node $s$. One approach is to identify $s$ from all the graph nodes, i.e., by setting $\mathcal{C} = \mathcal{N}$ in Eq. 1. However, this task is computationally intractable when the number of nodes is large, i.e., the model needs to find $s$ out of hundreds of thousands of nodes. Therefore, we sample some negative nodes $\{s_i^g\}_{i=1}^{n_{\text{neg}}}$ such that $(s_i^g, t) \notin \mathcal{E}$. Then, the candidate set to generate the source node becomes $\{s, s_1^g, \cdots, s_{n_{\text{neg}}}^g\}$ instead of all the graph nodes $\mathcal{N}$. We remark that such a sampling approach is standard for GNN pre-training and link prediction (Hamilton et al., 2017; Sun et al., 2020; Hu et al., 2020b).

◇ **Edge sampling for the edge discriminator.** In computing the loss for the discriminator, the number of edges in $\mathcal{E}_u$ is significantly larger than those in $\mathcal{E}_g$, i.e., we only mask a small proportion of the edges. To avoid the discriminator from outputting trivial predictions (i.e., all the edges belong to $\mathcal{E}_u$), we balance the two loss terms in $\mathcal{L}_d^e$. Specifically, we sample $\mathcal{E}_u^d \subset \mathcal{E}_u$ such that $|\mathcal{E}_u^d| = \alpha|\mathcal{E}_g|$, where $\alpha$ is a hyper-parameter. Then, we compute $\mathcal{L}_d^e$ on $\mathcal{E}_g$ and $\mathcal{E}_u^d$. Note that the node representations $h_d$ are still computed using all the generated and unmasked edges $\mathcal{E}_g$ and $\mathcal{E}_u$.

## 4.3 BASELINES

We compare our method with several baselines in the experiments. For fair comparison, all the methods are trained for the same number of GPU hours.

◇ **GAE** (Graph Auto-Encoder, Kipf & Welling 2016) adopts an auto-encoder for unsupervised learning on graphs. In GAE, node embeddings are learnt using a GNN, and we minimize the discrepancy between the original and the reconstructed adjacency matrix.

◇ **GraphSAGE** (Hamilton et al., 2017) encourages embeddings of neighboring nodes to be similar. For each node, the method learns a function that generates embeddings by sampling and aggregating features from the node's neighbors.

Table 1: Experimental results on homogeneous graphs. We report F1 averaged over 10 runs for the Reddit data and MRR over 10 runs for the product recommendation data. The best results are shown in **bold**.

|  | Reddit | Recomm. |
|---|---|---|
| w/o pre-train | 87.3 | 46.3 |
| GAE | 88.5 | 56.7 |
| GraphSAGE | 88.0 | 53.0 |
| DGI | 87.7 | 53.3 |
| GPT-GNN | 89.6 | 58.6 |
| GRACE | 89.0 | 51.5 |
| GraphCL | 88.6 | — |
| JOAOv2 | 89.1 | — |
| DiP-GNN | **90.7** | **60.1** |

Table 2: Experimental results on OAG-CS (heterogeneous). Left to right: paper-field, paper-venue, author-name-disambiguation. We report MRR over 10 runs. The best results are shown in **bold**.

|  | PF | PV | AD |
|---|---|---|---|
| w/o pre-train | 32.7 | 19.6 | 60.0 |
| GAE | 40.3 | 24.5 | 62.5 |
| GraphSAGE | 37.8 | 22.1 | 62.9 |
| DGI | 38.1 | 22.5 | 63.0 |
| GPT-GNN | 41.6 | 25.6 | 63.1 |
| GRACE | 38.0 | 21.5 | 62.0 |
| GraphCL | 38.0 | 22.0 | 61.5 |
| JOAOv2 | 38.6 | 23.5 | 62.8 |
| DiP-GNN | **44.1** | **27.7** | **65.6** |

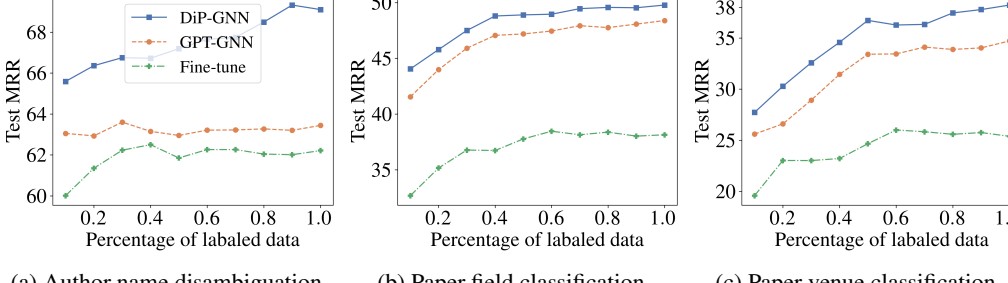

(a) Author name disambiguation.    (b) Paper field classification.    (c) Paper venue classification.

Figure 3: Model performance vs. amount of labeled data on OAG-CS.

◇ **DGI** (Deep Graph Infomax, Velickovic et al. 2019) maximizes mutual information between node representations and corresponding high-level summaries of graphs. Thus, a node's embedding summarizes a sub-graph centered around it.

◇ **GPT-GNN** (Hu et al., 2020b) adopts a generative pre-training objective. The method generates edges by minimizing a link prediction objective, and incorporates node features in the framework.

◇ **GRACE** (Graph Contrastive Representation, Zhu et al. 2020) leverages a contrastive objective. The algorithm generates two views of the same graph through node and feature corruption, and then maximize agreement of node representations in the two views.

◇ **GraphCL** (You et al., 2020) is another graph contrastive learning approach that adopts node and edge augmentation techniques, such as node dropping and edge perturbation.

◇ **JOAO** (Joint Augmentation Optimization, You et al. 2021) improves GraphCL by deigning a bi-level optimization objective to automatically and dynamically selects augmentation methods.

### 4.4 MAIN RESULTS

In Table 1 and Table 2, *w/o pre-train* means direct training on the fine-tuning dataset without pre-training. Results on the Reddit dataset are F1 scores averaged over 10 runs, and results on the product recommendation graph are MRR scores averaged over 10 runs. All the performance gain have passed a hypothesis test with p-value $< 0.05$.

Table 1 summarizes experimental results on the homogeneous graphs: Reddit and Recommendation. We see that pre-training indeed benefits downstream tasks. For example, performance of GNN improves by at $\geq 0.4$ F1 on Reddit (DGI) and $\geq 5.2$ MRR on Recommendation (GRACE). Also, notice that among the baselines, generative approaches (GAE and GPT-GNN) yield promising per-

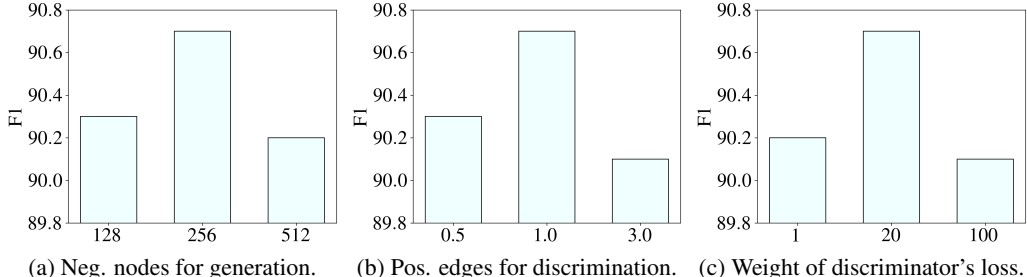

(a) Neg. nodes for generation.     (b) Pos. edges for discrimination.     (c) Weight of discriminator's loss.

Figure 4: Ablation experiments on Reddit. By default, we set the number of negative nodes to 256, the factor of positive edges to 1.0, and weight of the discriminator's loss to 20.

Table 3: Test F1 score of model variants on Reddit.

| Model | F1 |
|---|---|
| Edges+Features | 90.7 |
| Edges | 90.4 |
| Features | 90.2 |
| RandomEdges | 89.8 |

Table 4: Test F1 of models with different backbone graph neural networks on Reddit.

| Model | HGT | GAT |
|---|---|---|
| w/o pretrain | 87.3 | 86.4 |
| GPT-GNN | 89.6 | 87.5 |
| DiP-GNN | **90.7** | **88.5** |

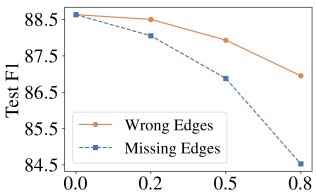

Figure 5: F1 vs. proportion of manipulated edges on Reddit.

formance. On the other hand, the contrastive method (GRACE, GraphCL and JOAO) does not scale well to large graphs, e.g., the OAG-CS graph which contains 1.1M nodes and 28.4M edges. By using the proposed discriminative pre-training framework, our method significantly outperforms all the baseline approaches. For example, DiP-GNN outperforms GPT-GNN by 1.1 on Reddit and 1.5 on Recommendation.

Experimental results on the heterogeneous OAG-CS dataset are summarized in Table 2. Similar to the homogeneous graphs, notice that pre-training improves model performance by large margins. For example, pre-training improves MRR by at least 5.1, 2.5 and 2.5 on the PF, PV and AD tasks, respectively. Moreover, by using the proposed training framework, models can learn better node embeddings and yield consistently better performance compared with all the baselines.

Recall that during fine-tuning on OAG-CS, we only use 10% of the labeled fine-tuning data (about 2% of the overall data). In Figure 3, we examine the effect of the amount of labeled data. We see that model performance improves when we increase the amount of labeled data. Also, notice that DiP-GNN consistently outperforms GPT-GNN in all the three tasks under all the settings.

### 4.5 ANALYSIS

◇ **Comparison with semi-supervised learning.** We compare DiP-GNN with a semi-supervised learning method: C&S (Correct&Smooth, Huang et al. 2020). Figure 6 summarizes the results. We see that C&S yields a 0.5 improvement compared with the supervised learning method (i.e., w/o pre-train). However, performance of C&S is significantly lower than both DiP-GNN and other pre-training methods such as GPT-GNN.

◇ **Hyper-parameters.** There are several hyper-parameters that we introduce in DiP-GNN: the number of negative nodes that are sampled for generating edges (Section 4.2); the number of positive edges that are sampled for the discriminator's task (Section 4.2); and the weight of the discriminator's loss

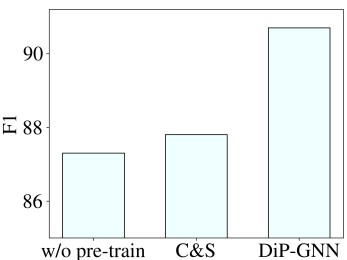

Figure 6: Test F1 on Reddit.

(Eq. 12). Figure 4 illustrate ablation experimental results on the Reddit dataset. From the results, we see that DiP-GNN is robust to these hyper-parameters. We remark that under all the settings, ours model behaves better than the best-performing baseline (89.6 for GPT-GNN).

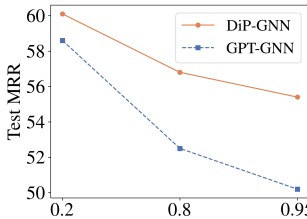

Figure 7: Performance vs. proportion of masked edges on product recommendation.

Table 5: Generator and discriminator performance vs. proportion of masked edges during pre-training. *Coverage* is the proportion of true edges input to the models.

| Masked% | Acc | | Coverage | | |
|---|---|---|---|---|---|
| | Gen. | Dis. | Gen. | Dis. | Ratio |
| 20 | 0.50 | 0.87 | 0.80 | 0.90 | ×1.13 |
| 80 | 0.33 | 0.84 | 0.20 | 0.46 | ×2.30 |
| 95 | 0.20 | 0.80 | 0.05 | 0.24 | ×4.80 |

◇ **Model variants.** We also examine variants of DiP-GNN. Recall that the generator and the discriminator operate on both edges and node features. We first check the contribution of these two factors. We also investigate the scenario where edges are randomly generated, and the discriminator still seeks to find the generated edges. Table 3 summarizes results on the Reddit dataset.

We see that by only using edges, model performance drops by 0.3; and by only using node features, performance drops by 0.5. This indicates that the graph structure plays a more important role in the proposed framework than the features. Also notice that performance of *RandomEdges* is unsatisfactory. This is because implausible edges are generated when using a random generator, making the discriminator's task significantly easier. We remark that performance of all the model variants is better than the best-performing baseline (89.6 for GPT-GNN).

Table 4 examines performance of our method and GPT-GNN using different backbone GNNs. Recall that by default, we use HGT (Hu et al., 2020c) as the backbone. We see that when GAT (Velickovic et al., 2018) is used, performance of DiP-GNN is still significantly better than GPT-GNN.

◇ **Missing edges hurt more than wrong edges.** In our pre-training framework, the generator is trained to reconstruct the masked graph, after which the reconstructed graph is fed to the discriminator. During this procedure, the graph input to the generator has *missing edges*, and the graph input to the discriminator has *wrong edges*. From Figure 5, we see that wrong edges hurt less than missing ones. For example, model performance drops by 0.7% when 50% of wrong edges are added to the original graph, and performance decreases by 1.8% when 50% of original edges are missing. This indicates that performance relies on the amount of original edges seen by the models. Intuitively, wrong edges add noise to the graph, but they do not affect information flow. On the contrary, missing edges cut information flow. Moreover, in practice we work with graph attention models, and the attention mechanism can alleviate the wrong edges by assigning low attention scores to them.

◇ **Why is discriminative pre-training better?** Figure 7 illustrates effect of the proportion of masked edges during pre-training. We see that when we increase the proportion from 0.2 to 0.8, performance of GPT-GNN drops by 6.1, whereas performance of DiP-GNN only drops by 3.3. This indicates that the generative pre-training method is more sensitive to the masking proportion.

Table 5 summarizes pre-training quality. First, the generative task (i.e., the generator) is more difficult than the discriminative task (i.e., the discriminator). For example, when we increase the proportion of masked edges from 20% to 80%, accuracy of the generator drops by 17% while accuracy of the discriminator only decreases by 3%. Second, the graph input to the discriminator better aligns with the original graph. For example, when 80% of the edges are masked, the discriminator sees 2.3 times more original edges than the generator. Therefore, the discriminative task is more advantageous because model quality relies on the number of observed original edges (Figure 5).

## 5 CONCLUSION AND DISCUSSIONS

We propose Discriminative Pre-Training of Graph Neural Networks (DiP-GNN), where we simultaneously train a generator and a discriminator. During pre-training, we mask out some edges in the graph, and a generator is trained to recover the masked edges. Subsequently, a discriminator seeks to distinguish the generated edges from the original ones. Our framework is more advantageous than generative pre-training for two reasons: the graph inputted to the discriminator better matches the original graph; and the discriminative pre-training task better aligns with downstream node classification. We conduct extensive experiments to validate the effectiveness of DiP-GNN.

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

## A  DETAILED ALGORITHM

Algorithm 1 is a detailed training pipeline of DiP-GNN. For graphs with vector features instead of text features, we can substitute the feature generation and discrimination modules with equations in Appendix B.

---

**Algorithm 1:** DiP-GNN: Discriminative Pre-training of Graph Neural Networks.

---

**Input:** Graph $\mathcal{G}_{\text{full}}$; edge masking ratio; feature masking ratio; number of negative samples for edge generator; proportion of positive samples for edge discriminator $\alpha$; weight of the discriminator's loss $\lambda$; number of training steps $T$.

**for** $t = 0, \cdots, T - 1$ **do**

    // Graph subsampling.

    Sample a subgraph $\mathcal{G} = (\mathcal{N}, \mathcal{E})$ from $\mathcal{G}_{\text{full}}$;

    // Edge generation.

    Initialize the generated edge set $\mathcal{E}_g = \{\}$ and the edge generation loss $\mathcal{L}_g^e = 0$;

    Construct the unmasked set of edges $\mathcal{E}_u$ and the masked set $\mathcal{E}_m$ such that $\mathcal{E} = \mathcal{E}_u \cup \mathcal{E}_m$;

    Compute node embeddings using $\mathcal{E}_u$;

    **for** $e = (n_1, n_2) \in \mathcal{E}_m$ **do**

        Construct candidate set $\mathcal{C}$ for $n_1$ ($n_2$ is given during generation) via negative sampling;

        Generate $\widehat{e} = (\widehat{n}_1, n_2)$ where $\widehat{n}_1 \in \mathcal{C}$;

        Update the generated edge set $\mathcal{E}_g \leftarrow \mathcal{E}_g \cup \{\widehat{e}\}$;

        Update the edge generation loss $\mathcal{L}_g^e$;

    // Text Feature generation.

    Initialize the feature generation loss $\mathcal{L}_g^f = 0$;

    **for** $n \in \mathcal{N}$ **do**

        For the node's text feature $\mathbf{x}_n$, mask out some of its tokens;

        Construct the generated text feature $\mathbf{x}_n^{\text{corr}}$ using the embedding of node $n$ (computed during edge generation) and the feature generation Transformer model;

        Update the feature generation loss $\mathcal{L}_g^f$;

    // Edge discrimination.

    Initialize the edge discrimination loss $\mathcal{L}_d^e = 0$;

    Compute node embeddings using $\mathcal{E}_g \cup \mathcal{E}_u$;

    Sample $\mathcal{E}_u^d \subset \mathcal{E}_u$ such that $|\mathcal{E}_u^d| = \alpha |\mathcal{E}_g|$;

    **for** $e = (n_1, n_2) \in \mathcal{E}_g \cup \mathcal{E}_u^d$ **do**

        Determine if $e$ is generated using the embedding of $n_1$ and $n_2$;

        Update the edge discrimination loss $\mathcal{L}_d^e$;

    // Text feature discrimination.

    Initialize the feature discrimination loss $\mathcal{L}_d^f = 0$;

    **for** $n \in \mathcal{N}$ **do**

        For the node's generated text feature $\mathbf{x}_n^{\text{corr}}$, determine whether each token is generated using the embedding of node $n$ (computed during edge discrimination) and the feature discrimination Transformer model;

        Update the feature discrimination loss $\mathcal{L}_d^f$;

    // Model updates.

    Compute $\mathcal{L} = (\mathcal{L}_g^e + \mathcal{L}_g^f) + \lambda(\mathcal{L}_d^e + \mathcal{L}_d^f)$ and update the model;

**Output:** Trained *discriminator* ready for fine-tuning.

---

## B  GENERATION AND DISCRIMINATION OF VECTOR FEATURES

Node features can be vectors instead of texts, e.g., the feature vector can contain topological information such as connectivity information. In this case both the generator and the discriminator are parameterized by a linear layer.

To generate feature vectors, we first randomly select some nodes $\mathcal{N}_g \subset \mathcal{N}$. For a node $n \in \mathcal{N}$, denote its feature vector $\mathbf{v}_n$, then the feature generation loss is

$$\mathcal{L}_g^f(W_g) = \sum_{n \in N_g} ||\widehat{\mathbf{v}}_n - \mathbf{v}_n||_2^2, \text{ where } \widehat{\mathbf{v}}_n = W_g^f h_g(n).$$

Here $h_g(n)$ is the representation of node $n$ and $W_g^f$ is a trainable weight. For a node $n \in \mathcal{N}$, we construct its corred feature $\mathbf{v}_n^{\text{corr}} = \widehat{\mathbf{v}}_n$ if $n \in \mathcal{N}_g$ and $\mathbf{v}_n^{\text{corr}} = \mathbf{v}_n$ if $n \in \mathcal{N} \setminus \mathcal{N}_g$.

The discriminator's goal is to differentiate the generated features from the original ones. Specifically, the prediction probability is

$$p(n \in \mathcal{N}_g) = \text{sigmoid}\left(W_d^d h_d(n)\right),$$

where $W_d^f$ is a trainable weight. We remark that the node representation $h_d(n)$ is computed based on the corred feature $\mathbf{v}_n^{\text{corr}}$. Correspondingly, the discriminator's loss is

$$\mathcal{L}_d^f(W_d) = \sum_{n \in \mathcal{N}} -\mathbf{1}\{n \in \mathcal{N}_g\} \log p(n \in \mathcal{N}_g) - \mathbf{1}\{n \in \mathcal{N} \setminus \mathcal{N}_g\} \log(1 - p(n \in \mathcal{N}_g)).$$

The vector feature loss $\mathcal{L}^f(\theta_g^e, W_g^f, \theta_d^e, W_d^f) = \mathcal{L}_g^f(\theta_g^e, W_g^f) + \mathcal{L}_d^f(\theta_d^e, W_d^f)$ is computed similar to the text feature loss.

## C IMPLEMENTATION AND TRAINING DETAILS

By default, we use Heterogeneous Graph Transformer (HGT, Hu et al. 2020c) as the backbone GNN. In the experiments, the edge generator and discriminator have the same architecture, where we set the hidden dimension to 400, the number of layers to 3, and the number of attention heads to 8. For the OAG dataset which contains text features, the feature generator and discriminator employs the same architecture: a 4 layer bi-directional Transformer model, similar to BERT (Devlin et al., 2019), where we set the embedding dimension to 128 and the hidden dimension of the feed-forward neural network to 512.

For pre-training, we mask out 20% of the edges and 20% of the features (for text features we mask out 20% of the tokens). We use AdamW (Loshchilov & Hutter, 2019) as the optimizer, where we set $\beta = (0.9, 0.999)$, $\epsilon = 10^{-8}$, the learning rate to 0.001 and the weight decay to 0.01. We adopt a dropout ratio of 0.2 and gradient norm clipping of 0.5. For graph subsampling, we set the depth to 6 and width to 128, the same setting as Hu et al. 2020b.

For fine-tuning, we use AdamW (Loshchilov & Hutter, 2019) as the optimizer, where we set $\beta = (0.9, 0.999)$, $\epsilon = 10^{-6}$, and we do not use weight decay. We use the same graph subsampling setting as pre-training. The other hyper-parameters are detailed in Table 6.

Table 6: Hyper-parameters for fine-tuning tasks.

| Dataset | Task | Steps | Dropout | Learning rate | Gradient clipping |
|---|---|---|---|---|---|
| Reddit | — | 2400 | 0.3 | 0.0015 | 0.5 |
| Recomm. | — | 1600 | 0.1 | 0.0010 | 0.5 |
| OAG-CS | PF | 1600 | 0.2 | 0.0010 | 0.5 |
| | PV | 1600 | 0.2 | 0.0005 | 0.5 |
| | AD | 1600 | 0.2 | 0.0005 | 0.5 |

