# OpenReview forum: "DiP-GNN: Discriminative Pre-Training of Graph Neural Networks"
_ICLR.cc/2023/Conference — Submitted to ICLR 2023_

### Official Review · Reviewer_K6Uv · 2022-10-23

**Confidence:** 4
**Correctness:** 3
**Technical Novelty And Significance:** 2
**Empirical Novelty And Significance:** 3
**Recommendation:** 5

**Clarity, Quality, Novelty And Reproducibility:**

Clarity: The statement of the paper is relatively clear and logical.

Quality: The presentation of the paper is relatively clearly.

Novelty: The authors propose a novel graph pre-training strategy which introduces the generative strategy. Compared with the current model, they fed the discriminator with the reconstructed graph to better match the original graph.

Reproducibility: The model framework is relatively easy to reproduce.


**Strength And Weaknesses:**

### Strength:
The authors introduce generative mechanism into graph pre-training and propose a DiP-GNN (Discriminative Pre-training of Graph Neural Network) framework. Meanwhile, compared with generative pre-training, the proposed reconstructed graph fed to the discriminator will better match the original graph.

### Weakness:
1) Lack of analysis about the generated graph. As shown in Fig. 1, the different masked-edge impacts the performance. Hope the authors can clarify how to choose the masked edge and analysis the influence of the masked edge.
2) Some statements are unclear and mislead the reader. For example, in section 1, the authors state that “we find that missing edges hurt more than wrong ones”, which is contrary to common sense but interesting. Although the authors utilize the experiment results to verify this point, I hope the author can explain and analyze it in more detail.
3) Lack of some explanation of equations. For example, the loss function in section 3 lacks an explanation and formula label.


**Summary Of The Paper:**

A popular pre-training GNN method is to mask out some edges and GNN is trained to recover them. However, there exists graph mismatch can mislead the performance. To handle this problem, the authors propose DiP-GNN which can better match the original graph. Specifically, the generator is trained to recover identities of the masked edges. Simultaneously, a discriminator is trained to distinguish the generated edges. Finally, the discriminator is chosen to accomplish the downstream tasks.

**Summary Of The Review:**

To handle the graph mismatch in graph generative pre-training, this paper proposes an interesting DiP-GNN (Discriminative Pre-training of Graph Neural Network) pre-training framework. Specifically, the generator is trained to recover identities of the masked edges. Simultaneously, a discriminator is trained to distinguish the generated edges.
The major strength is that the proposed reconstructed graph fed to the discriminator will better match the original graph, compared with generative pre-training.
The major weakness is that there lacks some detail explanation about the statement and the generated graph. Apart from that, some equations lack explanation. Meanwhile, some typo errors need to be rectified such as adding the equation label.
Apart from that, I have an interesting question. In section 3.1, the authors mask some edges not nodes. So, if all adjacent edges of the node are masked, how to handle this node in training.

---

> ### Author Response · Authors · 2022-11-12
> **Thank you for the comments**
>
> > Lack of analysis about the generated graph. As shown in Fig. 1, the different masked-edge impacts the performance. Hope the authors can clarify how to choose the masked edge and analysis the influence of the masked edge.
>
> We choose the masked edges randomly, and thus, only the masking ratio affects model performance. In Figure 7 and Table 5, we analyze the quality of the generator and the discriminator given different masking ratios. We explain the experimental results in the “Why is discriminative pre-training better?” section on page 9. In general, when we increase the masking ratio to an extreme extent (e.g., >80%), the quality of the generator deteriorates, while the quality of the discriminator decreases slower than the generator.
>
> > Some statements are unclear and mislead the reader. For example, in section 1, the authors state that “we find that missing edges hurt more than wrong ones”, which is contrary to common sense but interesting. Although the authors utilize the experiment results to verify this point, I hope the author can explain and analyze it in more detail.
>
> The intuition behind “missing edges hurt more than wrong ones” is the following. Wrong edges essentially add noise to the graph, but they do not affect information flow. On the contrary, missing edges cut information flow. Moreover, in practice we work with graph attention models, and the attention mechanism can alleviate the wrong edges by assigning low attention scores to them.
>
> > Lack of some explanation of equations. For example, the loss function in section 3 lacks an explanation and formula label.
>
> Thank you for the suggestion, we have fixed the issue.
>
> >  In section 3.1, the authors mask some edges not nodes. So, if all adjacent edges of the node are masked, how to handle this node in training.
>
> First, because we work with extremely large graphs, in practice we sample a small subgraph in each training iteration. Therefore, even though we may mask out all the edges of a node, the node’s information is still included in subsequent iterations. Second, if we have to ensure that the graph is still connected after masking, we can leave out at least one edge for each node.

---

### Official Review · Reviewer_S9Gb · 2022-10-23

**Confidence:** 4
**Correctness:** 3
**Technical Novelty And Significance:** 2
**Empirical Novelty And Significance:** 2
**Recommendation:** 5

**Clarity, Quality, Novelty And Reproducibility:**

The paper is well-written and easy to follow.

The novelty of the core idea/motivation seems a little ordinary though the details are different from the existing methods.

I have some questions about reproducibility mentioned in "Strength And Weakness" which requires the feedback of the authors.

**Strength And Weaknesses:**

### Strength
- The paper is well-written and easy to follow.
- The discussions of both edges and node features are sufficient.
- The method is proposed to be applied to the pre-training of GNN which is different from the existing GNN setings

### Weakness
- Although the authors claim that "*our setting is different from conventional self-supervised learning settings, namely we pre-train and fine-tune on two separate graphs*", I still have some questions about the experiments:
    - Could it show the superiority of DipGNN on transfer learning of GNN, since the authors claim " This meets the practical need of transfer learning"?
    - The compared methods seems a little confusing. If the competitors are mainly the self-supervised/unsupervised methods, the methods proposed by Kaveh Hassani et al. (ICML 2020) and SGC (Felix Wu et al. ICML 2019) are missing.
    - As DipGNN consists of pre-training and fine-tuning, do the all methods reported in Tables 1 and 2 have these two phases as well?
    - As the pre-training technique is usually used in large-scale datasets, some experiments on OGB datasets may be convincing.

- Although the details of generator and discriminator are different, the novelty seems a little common compared with the existing methods.

**Summary Of The Paper:**

In this paper, a pre-training method of GNN is proposed. The whole framework consists of generator and discriminator, and the discriminator is used as the pre-trained model. Therefore, the proposed method could be regarded as a variant of GAN on GNN, though there are significant differences between them. The authors also discuss the generation/discrimination of both edges and node features.

**Summary Of The Review:**

The paper is well-written and propose a method applied to the pre-training of GNN which is different from the existing GNN setings.

However, I have some concerns about the experimental settings and the originality.

---

> ### Author Response · Authors · 2022-11-12
> **Thank you for the comments**
>
> About experiments, we address the following:
>
>  * The experiments are under transfer learning settings, as stated in Section 4.1. For example, for the OAG dataset, we use paper before 2014 as pre-training nodes and paper from 2014-2016 as fine-tuning nodes. Note that our setting is different from transductive learning, in that we do not access the fine-tuning nodes during pre-training, and vice versa.
>
>  * The method MVGRL (Hassani et al.) is a contrastive learning method, which is outperformed by GraphSage [1] and DGI [2]. Our method outperforms both GraphSage and DGI. Moreover, we already included two graph contrastive learning baselines (GRACE and GraphCL) that are more recent than MVGRL. SGC is outperformed by GRACE by large margins (Table 1 in GRACE paper), and our method significantly outperforms GRACE. \
> [1] https://arxiv.org/abs/2106.02172 \
> [2] https://arxiv.org/abs/2108.10420
>
>  * Yes, all the methods have two phases.
>
>  * The adopted datasets have comparable sizes with OGB. We use OAG-CS, which contains 1.1M nodes. Our in-house product recommendation graph has an even larger scale, which contains over 2.7M nodes.
>
>
>
> About novelty, we address the following:
>
>  * We are the first to apply discriminative pre-training for GNNs, which empirically outperforms existing generative pre-training methods.
>
>  * Another contribution of our work is that we identify a graph mis-match issue: The input graphs in the pre-training stage mismatches the graphs in the fine-tuning stage due to missing edges. This is a novel observation for graph-related tasks, and our work deepens the understanding of graph neural networks. This observation is of independent interest, and can motivate follow-up discussions and works.

---

### Official Review · Reviewer_FMbA · 2022-10-31

**Confidence:** 4
**Clarity, Quality, Novelty And Reproducibility:** The paper is unclear in various aspec…
**Correctness:** 3
**Technical Novelty And Significance:** 3
**Empirical Novelty And Significance:** 3
**Recommendation:** 5

**Strength And Weaknesses:**

Strength
- The proposed method makes sense.
- The empirical results show the improved performance brought by the proposed methods on a number of  benchmarks.

Weaknesses
- There is no mentioning of n_1^hat or generated edges e_g in the loss of the generator. How should I understand that the generated edges are not captured in the generator loss?

- Also, it is said that the discriminator is to distinguish edges that are from the original graph and edges that are generated. It would make sense to contrast the masked edges vs the generated edges, especially the ones that are different. However, the training loss for the discriminator seems to still focus on the unmasked edges vs generated edges. I don’t quite understand the intuition.

- A masked edge has two end nodes n_1 and n_2. How do the authors decide which end node to predict?

- Why is the same lambda shared between edge loss and feature loss, given these two losses are so different?


**Summary Of The Paper:**

This paper proposes a discriminative method for pre-training Graph Neural Networks. The main idea is to simultaneously train a generator to recover identities of the masked edges, and train a discriminator to distinguish the generated edges from the original graph’s edges.


**Summary Of The Review:**

Overall, the proposed method makes sense. However, I feel the authors are selling the methodology wrong. Instead of saying that discriminative pre-training is better than generative training, the authors should present it as a method for applying generative and discriminative pre-training jointly. Otherwise, why don’t you just rely solely on discriminative pre-training?

---

> ### Author Response · Authors · 2022-11-12
> **Thank you for the valuable comments**
>
> I believe there are some misunderstandings about our algorithm. We summarize the proposed algorithm as the following:
>  1. For a given graph with edges $\mathcal{E}$, we first divide it into masked edges $\mathcal{E}_m$ and unmasked edges $\mathcal{E}_u$.
>  1. Then, a generator is trained using the unmasked edges $\mathcal{E}_u$ to recover the masked edges $\mathcal{E}_m$. The recovered (generated) set of edges is $\mathcal{E}_g$.
>  1. Because the generator cannot do perfect prediction, there are wrongly generated edges in $\mathcal{E}_g$, i.e., there are edges in $\mathcal{E}_g$ that do not belong to $\mathcal{E}$. Such edges are regarded as “fake” edges $\mathcal{E}_\text{fake}$ since they are not from the original graph. Correspondingly, the other edges (the unmasked edges $\mathcal{E}_u$ and the correctly generated edges in $\mathcal{E}_g$) are regarded as “true” edges $\mathcal{E}_\text{fake}$ since these edges are from the original graph.
>  1. Then, a discriminator is trained to distinguish the fake edges from the true edges.
>  1. Finally, we fine-tune the discriminator on downstream tasks.
>
> > There is no mentioning of n_1^hat or generated edges e_g in the loss of the generator. How should I understand that the generated edges are not captured in the generator loss?
>
> We remark that $\widehat{n}_1$ and $\mathcal{E}_g$ are used in the prediction process instead of the training process. That is, we first train a generator, and then we use the trained generator to generate (predict) the edge set $\mathcal{E}_g$. We have revised the paper to make this clearer.
>
> During the training process, for each edge $(n_1, n_2)$ in the masked set $\mathcal{E}_m$ (i.e., the set of edges we generate), we are given a node $n_2$, and we try to find $n_1$. This is included in the generator’s loss, namely we try to maximize $p(n_1|n_2)$. The idea is that if there exists an edge between $(n_1, n_2)$, then $p(n_1|n_2)$ should be large.
>
> > Also, it is said that the discriminator is to distinguish edges that are from the original graph and edges that are generated. It would make sense to contrast the masked edges vs the generated edges, especially the ones that are different. However, the training loss for the discriminator seems to still focus on the unmasked edges vs generated edges. I don’t quite understand the intuition.
>
> The intuition is not to distinguish “masked” edges and “generated” edges, but to distinguish the true edges (from the original graph) and fake edges (wrongly generated by the generator). We remark that we do not contrast the masked edges and the generated edges. This is because the generator is trained such that the generated edges should be close to the masked edges, i.e., the training objective of the generator is to recover the masked edges given the unmasked edges.
>
> To train the discriminator, the idea is for the model to tell whether an edge is true or fake (in or not in the original graph). Therefore, the training loss for the discriminator focuses on unmasked edges (which correspond to positive samples that are from the original graph) and wrongly generated edges (which correspond to negative samples).
>
> > A masked edge has two end nodes n_1 and n_2. How do the authors decide which end node to predict?
>
> Since we are working with undirected graphs, for an edge we randomly choose a node to predict.
>
> > Why is the same lambda shared between edge loss and feature loss, given these two losses are so different?
>
> There are 4 loss terms: edge loss for the generator, feature loss for the generator, edge loss for the discriminator, and feature loss for the discriminator. In practice, we find that adding one hyper-parameter (Eq. 3) suffices for a good performance.

---

> ### Author Response · Authors · 2022-11-12
> **Thank you for the comments**
>
> > Overall, the proposed method makes sense. However, I feel the authors are selling the methodology wrong. Instead of saying that discriminative pre-training is better than generative training, the authors should present it as a method for applying generative and discriminative pre-training jointly. Otherwise, why don’t you just rely solely on discriminative pre-training?
>
> In our method, the discriminator is the key component since we only fine-tune the discriminator in downstream tasks (we mentioned this in the abstract and in Section 3.3). We explained both intuitively and empirically why discriminative pre-training is better than generative pre-training.
>
> It is possible to rely solely on the discriminative task. One naive approach is that given a graph, we simply replace some edges with random edges and then apply discriminative training (see the paragraph “Model variants” on page 9). However, in practice we find that this task is too easy for the discriminator (see Table 3 “RandomEdges”): Without the generator, the discriminator has an accuracy >95%, whereas with the generator, the accuracy of the discriminator is around 85%. As suggested by existing works [1,2], we can improve the downstream fine-tuning if we properly increase the difficulty of the pre-training tasks. Therefore, instead of randomly generating edges, we train a generator to do so.
>
> [1] https://arxiv.org/abs/2202.08005 \
> [2] https://arxiv.org/abs/1909.11942

---

### Decision · Program_Chairs · 2023-01-20

**Decision:**

Reject

**Justification For Why Not Higher Score:**

The reviewers were quite clear about their concerns and generally did not feel enthusiastic about the paper, pointing out the lack of clarity and limited novelty.

**Justification For Why Not Lower Score:**

N/A

**Metareview: Summary, Strengths And Weaknesses:**

A popular pre-training GNN method is to mask out edges and train GNN to recover the masked edges. This paper proposes a discriminative method for GNN pre-training. The main idea is to simultaneously train a generator to recover masked edges and a discriminator to distinguish the generated edges from the original edges. The discriminator is used as the pre-trained model to accomplish the downstream tasks.

**Strengths:**

* Empirical results show the improved performance brought by the proposed method on many benchmarks.
* The new method can be regarded as a variant of GAN on GNN, though there are significant differences between them. The authors also discuss the generation/discrimination of both edges and node features.

**Weaknesses:**

* The paper is unclear in various aspects, as explained in the above comments. For example, it is unclear how to choose the masked edge and an analysis of the influence of masked edges was not fully provided.
* Reviewers have some concerns about the experimental settings and novelty.
* Lack of some explanation of equations and analyses of generated graphs.